# Perspectives on hiPSC-Derived Muscle Cells as Drug Discovery Models for Muscular Dystrophies

**DOI:** 10.3390/ijms22179630

**Published:** 2021-09-06

**Authors:** Elena Abati, Emanuele Sclarandi, Giacomo Pietro Comi, Valeria Parente, Stefania Corti

**Affiliations:** 1Neuroscience Section, Dino Ferrari Centre, Department of Pathophysiology and Transplantation (DEPT), University of Milan, 20122 Milan, Italy; elena.abati@unimi.it (E.A.); emanuele.sclarandi@gmail.com (E.S.); giacomo.comi@unimi.it (G.P.C.); 2Fondazione IRCCS Ca’ Granda—Ospedale Maggiore Policlinico, Neurology Unit, 20122 Milan, Italy; valeria.parente@policlinico.mi.it

**Keywords:** stem cell model, cellular differentiation, muscular dystrophy, iPSC, dystrophin, skeletal muscle, drug screening platforms

## Abstract

Muscular dystrophies are a heterogeneous group of inherited diseases characterized by the progressive degeneration and weakness of skeletal muscles, leading to disability and, often, premature death. To date, no effective therapies are available to halt or reverse the pathogenic process, and meaningful treatments are urgently needed. From this perspective, it is particularly important to establish reliable in vitro models of human muscle that allow the recapitulation of disease features as well as the screening of genetic and pharmacological therapies. We herein review and discuss advances in the development of in vitro muscle models obtained from human induced pluripotent stem cells, which appear to be capable of reproducing the lack of myofiber proteins as well as other specific pathological hallmarks, such as inflammation, fibrosis, and reduced muscle regenerative potential. In addition, these platforms have been used to assess genetic correction strategies such as gene silencing, gene transfer and genome editing with clustered regularly interspaced short palindromic repeats (CRISPR)/CRISPR-associated protein 9 (Cas9), as well as to evaluate novel small molecules aimed at ameliorating muscle degeneration. Furthermore, we discuss the challenges related to in vitro drug testing and provide a critical view of potential therapeutic developments to foster the future clinical translation of preclinical muscular dystrophy studies.

## 1. Introduction

Muscular dystrophies (MDs) are inherited disorders characterized by progressive skeletal muscle degeneration, clinically resulting in weakness in specific muscle groups, loss of ambulation, breathing and swallowing difficulties and, in most cases, reduced life span [1]. On a histopathological level, MDs share common “dystrophic” features, including muscle fiber degeneration, necrosis, replacement of muscle with connective and adipose tissues, inflammatory signs and reduced tissue regeneration capability impairing overall muscle structure and function [2]. These degenerative processes are induced by the lack or dysfunction of key myofiber proteins secondary to genetic mutations in their encoding genes [1].

MDs are characterized by the genetic and clinical heterogeneity of causative genes, pattern of inheritance, age of onset, rate of progression and type of muscle affected [1]. Cardiac and respiratory muscles are also frequently involved. To date, more than 50 genes have been implicated in up to 70 forms of MD [1]. The most common MDs are dystrophinopathies (Duchenne muscular dystrophy (DMD)) and Becker muscular dystrophy (BMD)) [3], myotonic dystrophies (DMs) [4,5], facioscapulohumeral muscular dystrophy (FSHD), and limb girdle muscular dystrophies (LGMDs) [6,7,8].

Dystrophinopathies are caused by mutations in the *DMD* gene (Xp21) that lead to the loss (DMD) of dystrophin or reduction (BMD) in its production. While thousands of pathogenic mutations have been described, most DMD-causing mutations are located in a “hotspot” region between exons 45 and 55 of the *DMD* gene, encoding the central rod domain of the protein [9]. Mutations in the *DMD* gene usually cause single- or multiexon deletions that alter the open reading frame (ORF) and insert a premature stop codon, thereby causing the synthesis of a nonfunctional and truncated dystrophin protein [10]. Conversely, the mutations that are responsible for BMD do not disrupt the ORF, ensuring the presence of residual dystrophin with variable levels and functionalities, which explains the relatively mild phenotype [11].

Dystrophin is a large muscle protein that is essential for preserving the integrity of the sarcolemma by anchoring the cytoskeleton to the extracellular matrix (ECM). On the membrane side, dystrophin is linked to the ECM through the dystrophin-associated glycoprotein complex (DGC), which includes sarcoglycan proteins (α-, β-, γ- and δ-SG) and dystroglycan (α- and β-DG) proteins located on the sarcolemma [1,7,12]. On the internal cytoplasmic side, dystrophin is anchored to the F-actin filaments of the cytoskeleton, establishing a mechanical link between myofibers and the ECM [12]. SG proteins also play a crucial role in stabilizing dystrophin, mainly through SG-DG binding [12]. Overall, the dystrophin complex stabilizes the membrane of the myofibers, protecting the muscle cell, in particular the sarcolemma, from damage by contractile forces [12]. Skeletal muscles lacking dystrophin or associated proteins are more subject to mechanical stress and damage, as excessive amounts of calcium enter cells during contraction, thereby inducing myofiber death [12]. Myofiber death, in turn, triggers skeletal muscle loss and connective tissue deposition, which are the main pathophysiological features of MDs [12]. As regards DMD treatment, several mutation-specific treatment options have become available for these patients in recent years. In fact, four Food and Drug Administration (FDA)-approved antisense oligonucleotides (ASOs) are currently available for DMD: Exondys 51 (eteplirsen), an exon 51-skipping ASO approved in 2016 [13]; Vyondys 53 (golodirsen) and Viltepso (viltolarsen), two exon 53-skipping ASOs approved in 2019 and in 2020, respectively [14,15]; and Amondys 45 (casimersen), an exon 45-skipping ASO approved in 2021 [16]. Moreover, ataluren, a small-molecule compound administered orally, has been available for the treatment of DMD caused by nonsense mutations since 2014. Ataluren indeed allows the bypass of nonsense mutations in messenger RNA (mRNA) and promotes the continuation of the translation process to produce a functioning protein [17]. However, this drug was only approved by the European Medical Agency (EMA) and not by the FDA. Regarding DMD, the standard of care also includes the use of immunosuppressants such as corticosteroids, which have been shown to postpone ambulation loss but have minimal impacts on the overall disease course [1].

Genetic hits on different proteins of the dystrophin complex or at other levels of the muscle contraction chain are responsible for other forms of MD. The dysfunction of the aforementioned SG proteins leads to sarcoglycanopathies, a LGMD subtype [8]. Similarly, dysferlinopathies are progressive muscle disorders that include LGMD R2 and Miyoshi myopathy (MM) [8] and are caused by mutations in the dysferlin (*DYSF*) gene, which disrupt muscle membrane resealing [8].

Myotonic dystrophy type 1 (DM1) is an autosomal dominant genetic multisystem disorder that affects many organs and systems, including muscle. It is characterized by a progressive loss of strength caused by the degeneration of muscle cells associated with muscle stiffness and impaired muscle relaxation (myotonia) [4]. DM1 is caused by expansion of a CTG trinucleotide repeat in the noncoding region of the dystrophia myotonica protein kinase (*DMPK*) gene. In patients with DM1, the 3′ untranslated region (UTR) of the *DMPK* gene contains more than 50 CTG repeats, while 5–37 CTG repeats are found in healthy individuals. The severity and age of onset of this disease are proportional to the repeat size.

In stark contrast with DMD, these MD subtypes can be treated with only symptomatic therapies, such as cardiovascular drugs, physical therapy, and noninvasive ventilation (NIV) devices.

It stands out that, currently, no therapies are available to counteract the pathogenic causes of most MDs. Recently, a variety of potential therapeutic strategies have been proposed, including novel small molecules and existing repurposed drugs, ASOs aimed at inducing mRNA modulation (in particular exon skipping) and gene silencing and gene editing techniques (Figure 1) [18,19,20,21]. However, these approaches have relatively limited efficacies, and preclinical validation and refinement are necessary to ensure maximal clinical effects. From this perspective, the development of effective disease models that can precisely recapitulate disease features is a key requisite. Here, we critically summarize the recent advances in skeletal muscle cell platforms, focusing in particular on new human in vitro models based on patient-specific human induced pluripotent stem cells (iPSCs) and their application for high-throughput drug screening and gene therapies for MDs.

## 2. Preclinical Models of MD

The quest for an effective disease model is one of the major hurdles that scientists face when searching for novel, translatable therapeutic strategies. Previous insights into MD pathogenesis were obtained from in vitro skeletal or cardiac muscle models based on immortalized myoblasts or cardiomyocytes (CMs) isolated from rodents (such as C2C12 and L6 cells) [22,23,24,25,26]. However, these models do not faithfully reproduce human disease.

Cultured primary myoblasts obtained from human skeletal muscle biopsies have been developed and provide a physiologically relevant model for processes such as muscle growth, atrophy, and regeneration as well as for metabolic studies. These cultures are derived from the in vitro proliferation of muscle satellite cells, adult tissue-specific stem cells located around the muscle fibers, under the basal lamina. When the biopsied muscle specimen is seeded in vitro, satellite cells proliferate as mononucleated myoblasts and can be differentiated into syncytial myotubes [27]. However, these cultures have limited proliferative capacity, particularly those from subjects with an MD genetic background, thereby diminishing the efficiency of this model. Other cell types obtained from muscles or blood were shown to have some myogenic potential in vitro, namely, the subpopulation of myogenic stem cells (CD133+), bone marrow/hematopoietic-derived stem cells, mesangioblasts, and pericytes, but their use as MD models is limited overall [28,29].

When considering CMs, reliable human primary cultures are even more difficult to obtain [30]. First, it is challenging to access adequate quantities of pre-mortem ventricular myocardial tissue. Then, isolated CMs are terminally differentiated and hold a limited life span, and they quickly undergo morphological and structural remodeling with a loss of potential disease hallmarks [30].

Embryonic stem cells (ESCs) are pluripotent cells that have an unlimited proliferative capacity and the ability to self-renew and differentiate into virtually all tissues of the three embryonic germ layers [31]. However, the use of human ESCs is controversial from an ethical point of view. In 2006, Takahashi and Yamanaka reached a fundamental scientific milestone by discovering a method for reprogramming adult somatic cells into pluripotent stem cells [32]. These cells were named induced pluripotent stem cells (iPSCs, human iPSCs (hiPSCs)) and are now widely used as models for different human diseases, including MDs [33,34]. hiPSCs advantageously harbor the natural MD mutation in the context of the whole human genomic background, which cannot be reproduced in animal models.

This technology overcomes some issues related to the use of primary cells as human MD models because it not only allows access to an unlimited number of cells but can also be used to study different types of muscle cells—both skeletal and cardiac cells—and also all the stages of muscle differentiation, including the early phases.

Knowledge of physiological skeletal muscle development enabled the establishment of several approaches for differentiating skeletal muscle cells from iPSCs (Figure 2) [33,34]. The key steps are represented by the activation of Wnt signaling and/or inhibition of bone morphogenetic protein (BMP). A myogenic phenotype can also be induced by the transgene overexpression of muscle-specific transcription factors, including PAX3, PAX7, and MyoD, and the transgene and small-molecule induction methods are often used in combination. We already reviewed the specific protocols for the differentiation of iPSCs into myogenic progenitors or myoblasts [34].

CMs may be used to model cardiovascular features of MDs, but they are among the most difficult cell type to differentiate from iPSCs. Signals induced by nodal, BMP and fibroblast growth factors (FGFs) are first required to generate cardiac mesoderm-like cells [35]. The resultant initial inhibition of the Wnt/β-catenin pathway is subsequently required for cardiac specification [35]. CMidentity and maturity then needs to be confirmed via functional tests assessing electrophysiology, contractility and/or calcium dynamics (Figure 2) [36].

The discovery of iPSCs paves the way for their use in translational research as a patient-specific tool for high-throughput drug screening and toxicity analyses as well as for the in vitro assessment of disease pathologies before the onset of typical secondary effects of inflammation. Thus, these cells replace primary cell cultures and immortalized myogenic cell lines.

## 3. hiPSCs for Drug Screening

### 3.1. Drug Screening for DMD

Before the advent of iPSCs, drug screening models were based on patient samples or immortalized myogenic human and rodent cell lines. However, neither of these models are ideal. While cells derived from humans can be used to directly model the effects of drugs on humans, their expansion capabilities are more limited than those of in vitro cell lines; immortalized cell lines may, in turn, exhibit genetic and metabolic anomalies. These drawbacks limit the ability of these models to faithfully simulate human diseases. iPSCs can overcome these constraints, providing a promising and valuable tool for both therapeutic purposes and basic scientific research (Table 1).

iPSCs were employed for drug screening purposes by Abujarour and colleagues (Table 1) [37]. They derived 16 iPSC lines from DMD and BMD patients and differentiated them by overexpressing the myogenic regulatory factor MyoD, obtaining myotubes. They then showed that these cells exhibited inflammatory-related gene alterations, including disrupted BMP/TGF signaling, and a diminished ability to fuse in myotubes, thus reproducing some key DMD/BMD features. Moreover, they demonstrated that iPSC-derived myoblasts could respond in terms of growth/maturation to the hypertrophic proteins insulin-like growth factor (IGF1) and wingless-type protein 7a (Wnt7a), as observed in primary myotubes. These results support potential therapeutic roles for Wnt7a and IGF1, which are now being investigated in preclinical and clinical DMD settings, respectively.

Later, Choi and colleagues established a human DMD model using iPSCs (Table 1) [38]. In their work, iPSC-derived DMD myoblasts presented disease-specific features, such as altered transcriptional profiles and intracellular signaling, as well as myotube defects. Additionally, they showed that DMD hallmarks could be partially rescued by genetic and pharmacological approaches based on BMP, TGF-β “dual SMAD” inhibitors.

Concomitantly, other researchers established an iPSC-derived DMD CM model of dilated cardiomyopathy that was used to test the ability of an amphiphilic surface copolymer, Poloxamer 188 (P188), to repair disrupted cellular plasma membranes (Table 1) [39]. The benefit of this therapeutic approach was confirmed for cardiac abnormalities linked to the lack of dystrophin, such as the increased permeability of the plasma membrane, calcium imbalance, and the hypercontraction of CMs resulting in cell death. Indeed, significant reductions in cytosolic Ca^2+^ levels and reductions in Caspase-3 activation and apoptosis were observed. These data support the idea that membrane fragility causes intracellular calcium overload even in the early stages of differentiation, suggesting the potential of membrane sealants in the treatment of DMD.

DMD CMs also exhibited decreased levels of endothelial and neuronal nitric oxide synthases (NOSs), with consequentially increased susceptibility to stress-induced injury [39]. Indeed, DMD CMs showed increased cell injury and death after 2 h of stress and recovery, and these phenomena were linked to upregulated levels of reactive oxygen species (ROS) and alteration of the mitochondrial membrane potential. Therefore, another group assessed the cardioprotective effect of nicorandil, a KATP (Kir6) channel opener and NO donor, on dystrophin-deficient iPSC-derived CMs by evaluating the lactose dehydrogenase (LDH) release and DNA damage in these cells pretreated with nicorandil via TUNEL staining (Table 1) [40]. The increased expression of SOD2, an antioxidant gene, and decreased mitochondrial ROS production were observed. In addition, hearts isolated from a murine model of DMD, *mdx* mice, showed functional recovery following perfusion with nicorandil. These results support the hypothesis that nicorandil protects against stress-induced cardiac injury and has potential as a new therapeutic agent for DMD cardiomyopathy.

A key aspect of drug screening is the scalability of the cell cultures. From this perspective, Uchimura and colleagues were able to myogenically differentiate human iPSCs by overexpressing tetracycline-inducible MyoD in a large-scale culture (384-well microplates) with a consistently high myotube differentiation efficiency, facilitating high-throughput drug screening [43].

Recently, a DMD hiPSC-derived myoblast drug screen identified compounds that improved the DMD phenotype in rodent models (Table 1) [41]. The researchers developed a DMD hiPSC-derived myoblast platform employing high-content imaging to detect characteristic morphological markers and myogenic proteins. They found that the defective fusion of myoblasts into myotubes was a recognizable pattern. Thus, they tested >1500 molecules from the Johns Hopkins Clinical Compound Library and selected those that enhanced the myogenic fusion of DMD cells, such as ginsenoside Rd, belonging to the FKT3 signaling pathway, and fenofibrate, linked to TGF-β signaling. Preclinical evaluations of *mdx* mice revealed that treatment with these two molecules significantly improved their pathological muscle features, supporting their therapeutic potential. Furthermore, fenofibrate mitigated the mitochondrion-induced programmed death of DMD CMs. These data indicate the feasibility of drug screening with hiPSC-derived DMD models.

### 3.2. Drug Screening for LGMDs

In 2019, Kokubu and colleagues established a drug screening platform based on myocytes from MM-derived iPSCs (Table 1) [42]. They identified nocodazole as an effective compound that was capable of increasing the cellular level of dysferlin, which, in turn, rescued membrane resealing following laser irradiation injury. The dysferlin upregulation was shown to be due to microtubule disorganization and involved autophagy rather than the proteasome degradation pathway. These data suggest that increasing the level of a deficient protein using small molecules has therapeutic potential for autosomal recessive LGMDs.

## 4. hiPSCs in the Development of Disease-Modifying Therapeutic Approaches for MDs

### 4.1. Antisense Oligonucleotides

#### 4.1.1. Antisense Oligonucleotides for DMD

Exon skipping represents a very promising therapeutic strategy for modifying molecular diseases. This approach, which to date has been applied mainly to DMD, aims to eliminate the “molecular defect” by directly modifying the dystrophin-encoding mRNA, correcting the genetic mutation that alters the ORF. This method allows the production of a shorter, less active but still functional protein, which appears to be capable of reducing the clinical manifestations of the disease and slowing ambulatory and pulmonary function decreases after long-term use [3].

Exon skipping is carried out using short fragments of RNA called ASOs, which target pre-mRNA sequences (Figure 1). Dystrophin-targeting ASOs have been tested in several animal models and patient-derived cell models, and they recently reached the clinical phase. As we mentioned above, four ASOs designed to skip exons 51 (eteplirsen), 53 (golodirsen and viltolarsen) and exon 45 (casimersen) of dystrophin mRNA seem to promote dystrophin restoration in DMD patients, and they have been conditionally approved by the FDA [13,14,15,16]. Limiting factors of this approach include the need for intrathecal injection due to the poor abilities of these drugs to cross the blood–brain barrier (BBB) and the brief half-lives of the drugs, resulting in the need for frequent administrations [3].

With regard to ASO testing on iPSC models, Dick and colleagues found that dystrophin was expressed in approximately 30% of iPSC-derived CMss (iPSC-CMs) harboring several mutations (exon 47–50 or 48–50 deletions) across the *DMD* gene after treatment with an ASO targeting exon 51 (Table 2) [44].

The efficacy of exon skipping was also tested in a DMD model based on iPSC-derived myotubes (Table 2).

In DMD, pathological changes result from the impairment of calcium influx following membrane damage. In vitro, the calcium influx in response to electric stimulation in DMD myocytes was excessive compared to that in the controls. Researchers observed a decrease in Ca^2+^ overflow and a reduction in creatin kinase (CK) secretion in DMD iPSC-derived myotubes in which dystrophin expression was restored by ASO transfection [45]. The researchers used an ASO that specifically skipped exon 45 to promote translation by linking exon 43 to exon 46 in the Δ44 patient and linking exon 44 to exon 48 in the Δ46–47 patient.

Overall, these data indicate the reliability of iPSC-based platforms as tools for evaluating the efficacy of exon-skipping drugs tailored to individual DMD patients.

#### 4.1.2. Antisense Oligonucleotides for DM1

DM1 iPSCs can be a useful tool for studying disease pathogenesis and for developing therapeutic strategies. At a preclinical level, hiPSCs have been used to screen candidate ASOs targeting RNA foci (Table 2). DM1 iPSC-derived myogenic cells were generated using a PAX7 conditional expression system, further differentiated into myogenic progenitors and then terminally differentiated myotubes [46]. Both DM1 myogenic progenitors and myotubes showed intranuclear RNA foci, resulting in the sequestration of muscleblind-like splicing regulator 1 (MBNL1). This model was used to screen ASOs, and a candidate ASO was identified that effectively abolished RNA foci and rescued mis-splicing in DM1 iPSC-derived myotubes.

### 4.2. Gene Transfer

#### 4.2.1. Gene Transfer for DMD

Research on gene transfer therapy began in the 1980s and is now starting to yield positive results in different diseases, emerging as one of the most promising therapeutic approaches for neuromuscular diseases (Figure 1). One striking example is spinal muscular atrophy (SMA) [68]. Indeed, the delivery of the *SMN1* gene (*SMN* cDNA) to infants with SMA type 1 using adeno-associated virus (AAV) serotype 9 dramatically improved neuromuscular function and prolonged life expectancy, making the case for approval of the drug, named Translarna (onasemnogene abeparvovec), by North American and European regulatory agencies [68].

Translational *DMD* gene transfer trials with AAV are ongoing. However, the efficiency of this approach is limited by the difficulty of packaging the large *DMD* gene (2.4 MB) into an AAV vector with a capacity of approximately 4.7 kB [69]. To overcome this obstacle, truncated forms of dystrophin, called microdystrophins, were developed [69]. Microdystrophins are dystrophin genes modified to only retain the essential parts of the protein to maintain maximal biological function while simultaneously reducing their length, and they have been proven to be safe and capable of inducing robust transgene expression in DMD patients [70].

AAV-based dystrophin delivery has not yet been tested in hiPSC models in vitro, but other vectors have already been screened. Kazuki and colleagues constructed a human artificial chromosome (HAC) vector carrying a complete genomic dystrophin sequence (HAC-DYS). HACs have several advantages as gene therapy vectors, including the ability to carry large genetic inserts, thus overcoming the size limitation of the dystrophin gene, at least in vitro [47]. HAC-DYS has been successfully transferred into DMD iPSCs via microcell-mediated chromosome transfer (MMCT) based on the fusion of target cells and microcells derived from chromosome donor cells. Subsequent analyses of iPSCs differentiated into teratomas showed dystrophin expression in muscle-like tissue. Likewise, HAC-DYS was detected in all tissues of chimeric mice derived from *mdx*-iPSCs.

Similarly, Choi and colleagues reported the partial rescue of DMD phenotypes by HAC-DYS gene transfer [38].

Therefore, the combination of patient-specific iPSCs and HAC-containing defective genes represents a powerful tool for gene therapy.

#### 4.2.2. Gene Transfer for LGMDs

LGMD R3 is caused by mutations in the gene encoding α-SG, a component of the dystrophin–glycoprotein complex connecting the F-actin cytoskeleton and the ECM [8]. The delivery of a class of wild-type vessel-associated stem cells, mesangioblasts, appeared to correct the dystrophic phenotype of the LGMD mouse model after injection into the femoral artery [48]. In this experimental setting, the authors differentiated iPSCs into mesangioblast-like mesodermal progenitor cells and genetically corrected them using a lentiviral vector carrying human α-SG cDNA under the muscle-specific myosin light chain 1F promoter and enhancer. The transgene was detected in the myotubes generated from these cells, thus introducing the possibility to correct mesangioblasts derived from LGMD R3 as well as the ability of these genetically corrected mesangioblasts to differentiate into myogenic cell lines and express the therapeutic transgene. Furthermore, muscle fibers expressing α-SG were detected in transplanted LGMD mice, and the treated animals showed significant functional amelioration of the dystrophic phenotype [48].

Regarding dysferlinopathies, Tanaka and colleagues described the efficient differentiation of MM-hiPSCs into mature myocytes via the inducible expression of MyoD1 [49]. They generated myotubes with defective membrane repair and rescued the phenotype by expressing full-length dysferlin through plasmid transfection.

These findings are extremely relevant, as they not only establish a model of recessive MDs but also pave the way for potential gene transfer therapeutics in the future.

### 4.3. Genome Editing

#### 4.3.1. Genome Editing for DMD

In recent years, the clustered regularly interspaced short palindromic repeats (CRISPR)/CRISPR-associated protein 9 (Cas9) endonuclease system has emerged as the most popular and powerful genetic tool for monogenic disorders such as DMD (Figure 1) [3]. CRISPR/Cas9 is indeed able to precisely edit specific DNA strands that are complementary to the CRISPR sequence.

CRISPR/Cas9 belongs to the vast family of CRISPR/Cas systems. Briefly, it includes three major types (I, II, III) and the less common type, IV [71]. The distinction between the CRISPR/Cas types is based on the respective signature genes and the typical loci organization. The signature gene for type I system is *cas3*, which encodes a large protein with a helicase possessing a single-stranded DNA (ssDNA)-stimulated ATPase activity. The signature gene for type II CRISPR/Cas systems is *cas9*, which encodes a multidomain protein that combines all the functions of effector complexes and the target DNA cleavage and is essential for the maturation of the crRNA. Type III systems possess the signature gene *cas10*. CRISPR/Cas systems and subtypes were thoroughly discussed in [71].

The application of CRISPR/Cas9 to hiPSCs results in the restoration of DNA double-stranded breaks (DSBs) via both homology-directed repair (HDR) and nonhomologous end joining (NHEJ). In HDR, the DSB is repaired by the precise introduction of a DNA template, a single-stranded oligodeoxynucleotide (ssODN) or a double-stranded DNA sequence, into the host genome. In contrast, NHEJ is error susceptible and results in small insertions or deletions (INDELs) of nucleotides at the break site, which are selected depending on the patient’s mutation and the editing goal.

One potential goal is exon skipping, especially since the SpCas9 protospacer adjacent motif (PAM) sequence is identical to exon splicing consensus regions (NGG or NAG), exon deletion, exon reframing or base modification.

In particular, exon skipping and exon deletion corrections can be made by employing single-guide RNAs (sgRNAs) to insert large INDELs via a “single cut” at a splice acceptor site (SAS) or by employing two sgRNAs to create a “double cut” and remove one or more mutation-clustering hotspot exons, such as exons 45–55, in which 47% of the mutations accumulate [56].

Depending on the editing goal, the genomic correction can yield a full-length preserved gene, as occurs with duplication and pseudoexon mutations, or a shortened but functional dystrophin gene, resembling the genetic features of BMD. Nevertheless, both outcomes can lead to the “reframing” of the proper dystrophin transcript ORF.

This technique is advantageously useful for endogenous genes, unlike gene transfer, and permanent, unlike ASOs. A critical issue regarding the CRISPR/Cas9 system that must be addressed is its potentially low specificity, manifesting as off-target effects, which could result in unexpected mutations in the genome. However, recent studies have reported low levels of off-target gene expression in tissues [72]. In addition, engineered versions of Cas9 have been developed in order to improve its specificity, such as nickase Cas9 (Cas9n), enhanced specificity Cas9, and high-fidelity Cas9 [73].

While this issue could potentially affect its clinical applicability, the AAV vector delivery of CRISPR/Cas9 in vivo is considered a promising tool for the clinical translation of gene editing therapies. Thus, showing proof-of-concept for proper mutation correction in vitro might result in the rapid translation of this approach.

CRISPR/Cas9 allows for precise endogenous genomic correction in cultured cells, including hiPSCs (Table 2). Genetically corrected hiPSC lines can provide autologous healthy cell sources, which are useful as both effective personalized disease models to assess the therapeutic effects of candidate treatments and, potentially, as regenerative therapies in the context of cell transplantation strategies.

Li and colleagues analyzed the genetic correction approach in more closely DMD patient-derived iPSCs using both the CRISPR/Cas9 and TALEN platforms, comparing three different correction methods: exon skipping, frameshifting, and exon knock-in, and subsequently performed a genome-wide analysis of off-target mutations [50]. The results show that exon knock-in was superior at restoring full-length dystrophin with minimal off-target mutagenesis. Moreover, the authors developed a k-mer approach that, in contrast with other programs, provided a whole-genome view of targetable regions and limited off-target mutations.

Later, Young and colleagues demonstrated the ability of a single CRISPR/Cas9 pair to promote the permanent skipping of exons 45–55 in the human *DMD* gene via the deletion and NHEJ methods to thereby reframe the gene in iPSC-derived skeletal muscle cells and CMs [51]. Reframed CMs and skeletal muscle myotubes showed improved membrane integrity and restored dystrophin expression. The authors tested this approach in vivo by injecting reframed skeletal muscle cells into the tibialis anterior muscles of NSG-*mdx* mice, and they observed successful rescue of the dystrophin–glycoprotein complex. The proposed *DMD* genomic correction is clinically relevant since it is applicable in approximately 60% of the mutations in DMD patients. The researchers pointed out the significant therapeutic potential of a single pair of gRNAs to treat a large number of DMD patients.

Another group of researchers developed a new CRISPR/Cas protein called CRISPR/Cpf1, a class 2 CRISPR effector that functions as an RNA-guided endonuclease and is capable of genome editing in a manner similar to that of Cas9 [52]. This strategy has been exploited to address mutations in DMD patient-derived iPSCs by either exon skipping or nonsense mutation correction. Dystrophin expression was restored after the cells were differentiated into CMs, resulting in contractile function improvement. The efficiency of Cpf1-mediated *DMD* correction in human cells was also demonstrated in a DMD mouse model, in which dystrophy hallmarks were reverted following Cpf1-mediated germline editing, resulting in enhanced muscle contractility.

The therapeutic efficacy of genetic iPSC-CM correction was further tested by using these cells to derive a three-dimensional (3D) engineered heart muscle (EHM) as a model of dilated cardiomyopathy and correcting *DMD* mutations via the CRISPR/Cas9 method with sgRNAs [54]. The results were promising, as EHM contractile function was restored. In particular, correcting 30% of the CMs resulted in partial contractility improvement, while functionality was completely recovered when 50% of the CMs were corrected. Different sgRNAs were tested to identify the optimal sgRNA capable of modifying conserved RNA splice sites in 12 exons in the hotspot region of the *DMD* gene.

Exons 2–8 of the *DMD* gene, encoding amino-terminal actin-binding domain 1 (ABD-1), represent a mutational hotspot. Kyrychenko and colleagues developed three different approaches for correcting mutations in the ABD-1 region of the *DMD* gene by using CRISPR/Cas9 to remove exons 3–9, 6–9, or 7–11 in iPSCs and by evaluating the phenotype of iPSC-derived CMs [53]. All three approaches allowed the expression of a truncated dystrophin protein and the rescue of CM features (contractility and calcium currents), but the deletion of exons 3–9 by genomic editing was especially effective at correcting disease-causing ABD-1 mutations.

In addition, the disruption of the splice acceptor site in exon 45 by the plasmid or mRNA delivery of CRISPR/Cas9 to DMD iPSCs with an exon 44 deletion successfully restored dystrophin protein expression in differentiated myoblasts [55].

The CRISPR/Cas9 strategy was applied successfully in a pig model of DMD lacking exon 52 as well as in patient-derived iPSCs with the same mutation [58]. To overcome the AAV size limitation, Cas9 and other proteins were split into two smaller parts that reconstituted themselves when coexpressed via an intein-mediated trans-splicing method. Thus, an AAV9 carrying an intein-split Cas9 and two gRNAs targeting sequences flanking exon 51 (AAV9-Cas9-gE51) were injected into the DMD pig, leading to the restoration of a shortened dystrophin (DMDΔ51–52) and the amelioration of skeletal muscle, cardiac and systemic features. Indeed, in human-derived myoblasts and CMs of a patient lacking DMDΔ52, the AAV6-Cas9-g51-mediated deletion of exon 51 promoted the expression of dystrophin and rescued the diseased cellular phenotype.

Continuous expression of the CRISPR/Cas9 nuclease and gRNA from viral vectors may induce off-target mutagenesis and immunological reactions. From this perspective, transient delivery might be useful for therapeutic genome editing correction. Gee and colleagues established a new method of delivery based on an extracellular nanovesicle ribonucleoprotein system called NanoMEDIC [59]. Chemical dimerization leads to the inclusion of the Cas9 protein into extracellular nanovesicles; afterwards, a viral RNA packaging signal and two self-cleaving riboswitches tether and localize sgRNA into nanovesicles. In this way, they performed genome editing with a high transfection rate in different cell types, including hiPSCs, neurons, and myoblasts. NanoMEDIC was also shown to induce the excision of 90% of exon 45 in skeletal muscle cells derived from DMD iPSCs. NanoMEDIC also induced exon skipping in vivo in *mdx* mice upon intramuscular injection.

The precise correction of *DMD* exon deletion mutations can be achieved by base and prime editing. CRISPR technology now has the potential to precisely edit individual nucleotides, thereby allowing the correction of genetic mutations. In addition, base editors can also be exploited to target splice motifs, eliciting exon skipping. In this regard, a cytosine base editor (CBE) was employed at different intronic motifs to alter the splicing of different genes, including the *DMD* gene, in DMD patient-derived iPSCs [60]. However, this tool might have off-target effects on DNA and RNA [18].

To overcome this limitation, an adenine base editor (ABE) was applied to modify splice donor sites on the *DMD* gene and to correct the deletion of exon 51 in CMs derived from hiPSCs, thereby restoring dystrophin expression [61]. The researchers reported multiple successful CRISPR/Cas9 nucleotide gene editing strategies for skipping the errant “stop” signal, restoring up to 97% protein production. The same system was tested in vivo on a DMD mouse model, as ABE was packaged into AAV9 using a split-intein system, thereby restoring dystrophin protein expression [61].

To increase the efficacy and reduce the off-target effects, Morisaka and colleagues substituted CRISPR/Cas9, belonging to the Class 2 CRISPR system, with CRISPR/Cas3, belonging to the Class 1 CRISPR system, which induced broad and unidirectional genome editing in human cells [57]. They tested this alternative tool in DMD iPSCs promoting the skipping of exon 45. Given the long intervals between *DMD* exons, this could serve as a good proof-of-principle for the therapeutic application of long-range deletions induced by the Cas3 system. Even if the skipping and restoration of dystrophin are achievable, the CRISPR/Cas3 system has the following limitations: (1) a lower editing efficiency than the CRISPR/Cas9 system; (2) difficulty delivering multiple effectors to cells or tissues; and (3) immunogenic potential.

Nonetheless, it could prove to be useful for specific sequences for which the onsite design of sgRNAs is difficult, such as those far from PAM sequences, repetitive sequences, and transposon elements.

In summary, among the different methods of gene editing that have been applied to correct *DMD* mutations, the CRISPR/Cas9 system stands out as the most promising and efficient patient-specific approach. Other CRISPR systems, such as Class I CRISPR systems, although in their infancy, could be exploited in the future for specific mutations. These data provide proof-of-principle for the correction of diverse *DMD* mutations by nucleotide editing with minimal genomic modifications. This strategy can advantageously be used to correct multiple mutations with the same validated approach. Therefore, the optimization of delivery methods will permit increased genomic editing in DMD patients in vivo.

#### 4.3.2. Genome Editing for Autosomal Recessive Muscular Dystrophies

Regarding autosomal recessive LGMDs, Turan and colleagues reported the in situ correction of the mutated dysferlin gene in LGMD R2 and of the mutated SGCA gene in LGMD R3, confirming the validity of using disease-specific iPSC models to assess the correction approach (Table 2) [62]. The authors successfully used the CRISPR/Cas9 gene editing system to obtain both the dysferlin and α-SG proteins at the correct locations following the correction of single-point mutations in the respective genes using short single-stranded oligonucleotides for HDR enhanced by a site-specific DSB in patient-derived iPSCs. This approach led to the accurate and smooth correction of the specific alleles. Despite the fact that the efficiency was reduced compared to those of other targeting approaches, this strategy was still capable of correcting the expression of missing proteins. To make this approach more versatile and applicable in the clinic, the authors developed a method based on the insertion of wild-type genes into safe harbors such as Hipp11 (H11), a locus identified for efficient transgene knock-in and expression, and the adeno-associated virus integration site 1 (AAVS1) locus using dual integrase-assisted exchange (DICE) or TALEN/CRISPR/Cas9-assisted homologous recombination.

The CRISPR/Cas9 approach was also used to correct LGMD R1 iPSCs, demonstrating the rescue of the calpain 3 gene in myotubes in vitro (Table 2) [63]. The transplantation of genomic-corrected LGMD R1 myogenic cells into a mouse model that was both immune- and CAPN3-deficient resulted in muscle engraftment and the rescue of CAPN3 expression.

These results serve as proof-of-concept for the combination of genome editing and iPSC technologies to foster autologous cell therapies for LGMDs.

#### 4.3.3. Genome Editing for Autosomal Dominant Muscular Dystrophies

Notably, genome editing can potentially be used to permanently correct the defective gene in DM1 (Table 2). Xia and colleagues proposed a strategy for preventing the transcription of the CTG expansion repeats rather than editing the repeat itself [74]. They introduced premature polyadenylation (poly(A)) signals between the stop codon and the repeat in exon 15 to induce the premature termination of RNA synthesis. Corrected iPSCs and their derivatives (neuronal and cardiac cells) were clear of nuclear foci, and splicing abnormalities were rescued. This interesting strategy not only eliminates the synthesis of toxic repeats in RNA but also allows for the transcription of *DMPK* with a novel artificial 3′ UTR in the cells. In this way, *DMPK* haploinsufficiency may be prevented. Possible limitations include the fact that the novel 3′ UTR may lead to abnormal regulatory features of *DMPK* transcripts and that pathogenic repeats are still present in the genome, which may negatively affect the replication and heterochromatinization of the locus. Furthermore, the persistence of (CUG)n repeat transcription may result in RAN translation. The authors hypothesized that corrected DM1 hiPSC-derived skeletal muscle progenitor cells can be used as a cell source for transplantation.

Other studies demonstrated that removal of the CTG expansion repeats from the 3′- or 5′-UTR of the *DMPK* gene with dual sgRNA/CRISPR/Cas9 prevented nuclear focus formation and splicing alterations, reverting DM1 iPSCs to a normal phenotype [65,66,67]. Southern blot and single-molecule real-time (SMRT) sequencing demonstrated the correct excision.

However, doubts exist regarding the DNA-directed strategy despite the promising results obtained by these groups in terms of *DMPK* gene normalization. Xia and colleagues frequently observed the inversion of flanked CTG repeats, although these results are unpublished [64]. Such findings suggest the feasibility of using this approach to obtain isogenic cell models with pure deletions for research purposes but not for in vivo therapies.

Despite potential concerns and the need for further research, these studies potentially validate the use of CRISPR/Cas9 for gene editing in subjects with dominant MDs.

## 5. Challenging Issues and Future Perspectives Regarding iPSC Models and Therapeutic Screening

In the last decade, iPSC-based disease models have offered unprecedented possibilities for establishing high-throughput drug discovery and safety test platforms. Recently, even more opportunities have been provided by 3D organoids and personalized organs-on-chips in association with genome editing, high-throughput imaging and omics methods and the incorporation of artificial intelligence data analysis methods.

The major critical aspect of disease modeling is the ability of the platform to faithfully recapitulate the relevant pathophysiology and neuropathological features. Demonstrating the sufficient quality and differentiative capacity of iPSCs is a prerequisite for high-throughput phenotypic analyses.

Generating iPSCs from somatic cells remains one of the challenges associated with iPSC-based models. The selection of the cellular reprogramming method is of crucial importance, as it can influence the final MD phenotype observed in vitro. The first key step is the selection of the donor cell type, usually dermal fibroblasts from punch skin biopsies or mononuclear cells from peripheral blood [75]. Research has shown that the residual epigenetic memory of the originating somatic cell source is passed on to derived iPSCs, thus affecting their differential potential, which is influenced by the parental cells. However, this residual epigenetic memory is reduced throughout the cell culture period [76].

Concerning iPSC reprogramming, the initial methods were based on the use of retroviral and lentiviral vectors to deliver reprogramming factors. This entailed the risk of inactivating tumor suppressor genes and/or activating oncogenes due to insertions at specific sequences as well as the risk of iPSC alterations due to the constitutive expression of reprogramming factors. To circumvent this contingency, transient, integration-free methods have been developed to deliver reprogramming factors. Among them, viral delivery and transient transfection methods using Sendai virus, adenovirus, episomal plasmids and synthetically modified mRNAs have been the most widely used [75]. This strategy reduces the risk of phenotype modifications resulting from insertional mutagenesis.

Throughout the in vitro culture period, iPSCs can acquire chromosomal abnormalities, genetic instability, copy number variants and loss of heterozygosity, and these phenomena need to be evaluated and minimized.

Often, iPSC-based disease modeling studies include the comparative analysis of one or a few patient-derived iPSC lines with their respective sex-matched controls (or recently, with their respective isogenic cell lines created with gene editing) to evaluate the pathological phenotype. These results can be biased by the genomic background of the subject or by the features of the single clone; therefore, the use of relatively large cohorts of patient-derived iPSCs is necessary for the acquisition of robust in vitro data.

The acquisition of a precise and mature phenotype represents another challenge regarding iPSCs. iPSC-derived cells often exhibit immature features similar to their respective embryonic or fetal phenotypic cells or harbor a heterogeneous mixture of phenotypic subtypes at the end of the differentiation protocol. This can cause analytical variation and make analyzing the late-onset phenotype difficult.

Moreover, muscles comprise a complex system that includes cranial, trunk, and limb muscles, each of which have different developmental origins and programs. Muscles themselves are composed of slow or fast myofibers, characterized by different types of myosin heavy chain genes. This complexity makes it considerably challenging to faithfully mirror muscle pathophysiology in vitro. Indeed, muscle innervation is needed to carefully reproduce the phenotype. Late-onset disease phenotypes represent another obstacle, as they are difficult to reproduce in culture. In the future, the development of 3D cultures such as organoids derived from iPSCs might provide an effective platform to study diseases at the tissue or organ level, including spinal cord/muscle innervation, even for a long period [77]. The so-called “muscle-on-chip” method, representing a 3D skeletal muscle tissue grown within a microfluidic device, is another recent innovation that can recapitulate the architectural and structural complexities of muscle and thus emerge as a new complementary tool for drug testing [78]. In addition, preconditioning and cell engineering strategies could be exploited to enhance the clinical application of iPSC-based therapies [79].

Altogether, these findings suggest that iPSC technology will become increasingly relevant for muscle disease modeling and therapeutic development in the future.

## 6. Conclusions

The development and optimization of iPSC-based muscle platforms as representative models for human studies have allowed scientists to unravel disease-specific mechanisms. Further advances may finally facilitate the complete elucidation of key pathogenetic events and thereby lead to the development of effective treatments for muscular dystrophies. iPSC models can provide a useful platform for evaluating the efficacy/side effects of therapeutic technologies, such as small molecules, ASOs, gene transfer, and CRISPR/Cas9 genome editing. In addition, genetically corrected lines could represent a valuable cell source for autologous ex vivo gene therapies for the regenerative treatment of MDs. The increasing capacity to manipulate genomes has the potential to accelerate the translation of new therapeutic strategies from in vitro models to clinical settings.

Overall, iPSC technology, combined with gene silencing, transfer and editing techniques, is potentially revolutionary and might bridge the gap between preclinical and clinical trials, enabling more reliable testing models for precision medicine and drug discovery.

## Figures and Tables

**Figure 1 ijms-22-09630-f001:**
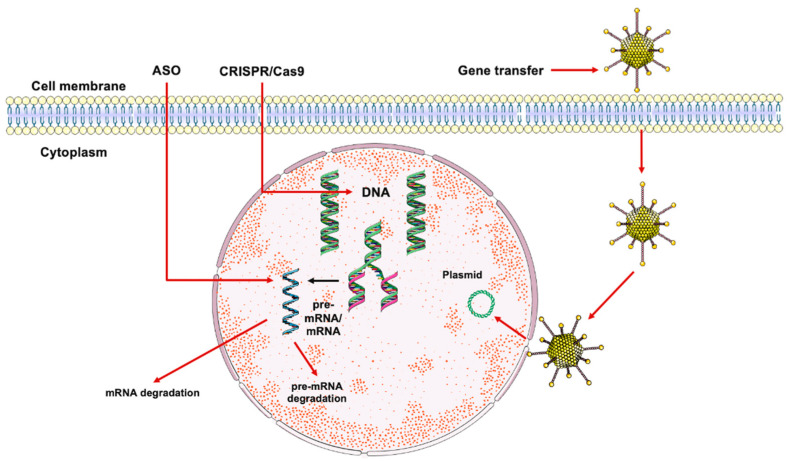
Endogenous pathways may be exploited to achieve therapeutic gene manipulation. Antisense oligonucleotides (ASOs) complementary to nuclear pre-mRNAs or cytoplasmic mRNAs exploit the RNAse-mediated pathway, binding to complementary sequences to induce their degradation. This process may occur both in the nucleus and in the cytoplasm. Clustered regularly interspaced short palindromic repeats (CRISPR)/CRISPR-associated protein 9 (Cas9) acts on double-stranded DNA, causing breakage at target sites in the genome. Gene transfer is achieved via viral vectors, usually an adeno-associated virus or a lentivirus, that are used to carry complementary DNA sequences into the cells, to induce the expression of one or more particular genes. Figure modified from images from Servier Medical Art, licensed under a Creative Common Attribution 3.0 Generic License. http://smart.servier.com/, accessed on 25 July 2021.

**Figure 2 ijms-22-09630-f002:**
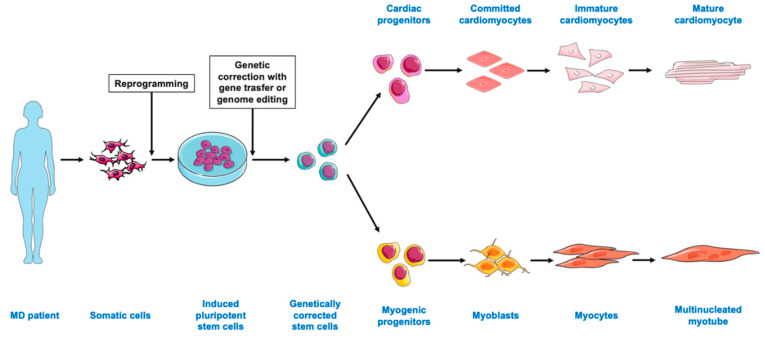
iPSC applications for muscular dystrophies (MDs). Somatic cells obtained from the peripheral blood or skin biopsies of MD patients can be reprogrammed into iPSCs. iPSCs can be genetically corrected in vitro via gene transfer or genome editing methods. The first step is then to differentiate iPSCs into mesodermal progenitors. Mesodermal progenitors may then commit to a cardiac lineage fate, differentiating into immature cardiomyocytes, which can then undergo proper maturation. Mesodermal progenitors may also be specified into myoblasts, which will undergo a skeletal muscle lineage differentiation—myoblasts, myocytes and then multinucleated myotubes. Mature cardiomyocytes and myotubes can then be used for disease modeling, drug screening and as a cell source for transplantation. Figure modified from images from Servier Medical Art, licensed under a Creative Common Attribution 3.0 Generic License. http://smart.servier.com/, accessed on 25 July 2021.

**Table 1 ijms-22-09630-t001:** iPSC-based models of muscular dystrophies for drug screening.

Disease	Cell Type Obtained	Differentiation Strategy	Screened Molecule(s)	Results	References
DMD, BMD	Myotubes	Overexpression of the myogenic regulatory factor MyoD	Insulin-like growth factor (IGF1) and wingless-type protein 7a (Wnt7a)	Increase in myotube diameter	[37]
DMD	Myoblasts	Exogenous growth factor-based protocol	Pharmacological “dual-SMAD” inhibition (LDN193189 and SB431542)	Reversal of increased nuclear localization of pSMAD protein and expression of IL-6 and -8 and Col3; rescue of myoblasts fusion defects	[38]
DMD	Cardiomyocytes	Exogenous growth factor-based protocol	Poloxamer 188 (P188)	Decreased resting cytosolic Ca^2+^ level, repressed caspase-3 activation	[39]
DMD	Cardiomyocytes	Exogenous growth factor-based protocol	Nicorandil	Expression of SOD2 and decreased mitochondrial ROS production	[40]
DMD	Myoblasts	Exogenous growth factor-based protocol with FACS purification of NCAM+/HNK1− cells	Ginsenoside Rd, and fenofibrate	Enhancement of myogenic fusion	[41]
MM	Myocytes	Exogenous growth factor-based protocol	Nocodazole	Increased dysferlin level and rescue of membrane resealing properties after injury	[42]

Abbreviations: BMD, Becker muscular dystrophy; DMD, Duchenne muscular dystrophy; FACS, fluorescence-activated cell sorting; IL, interleukin; iPSC, induced pluripotent stem cell; MM, Miyoshi myopathy; ROS, reactive oxygen species.

**Table 2 ijms-22-09630-t002:** Summary of key studies using iPSCs for gene silencing, transfer or editing.

Intervention	Cell Type	Outcome	References
Antisense oligonucleotides (ASOs)			
ASO-mediated *DMD* exon 51 skipping	DMD iPSC-derived CMs	Restoration of dystrophin expression in 30% of CMs.	[44]
ASO-mediated *DMD* exon 45 skipping	DMD iPSC-derived myotubes	Restoration of dystrophin expression, decrease in Ca^2+^ overflow and reduction in CK secretion.	[45]
2′-OMe-PT-(CAG)7-ASO to abolish RNA foci	DM1 iPSC-derived myogenic cells	Reduction in the number of nuclei containing RNA foci and increased inclusion of *BIN1* exon 11.	[46]
Gene transfer			
Human artificial chromosome vector carrying a genomic dystrophin sequence (HAC-DYS)	DMD iPSCs	Restoration of dystrophin expression in muscle-like tissues upon differentiation of iPSCs into teratomas.	[47]
Human artificial chromosome vector carrying a genomic dystrophin sequence (HAC-DYS)	DMD iPSCs	Restoration of dystrophin expression and reversal of the pathologic phenotype (reduced nuclear localization of phosphorylated SMAD and reduced IL-6, IL-8 and collagen 3 expression compared to those in untreated cells).	[38]
Lentiviral vector-encoding human α-sarcoglycan cDNA	LGMD R3 iPSCs	Restoration of alpha-SG expression, functional amelioration upon transplantation into α-SG-null mice.	[48]
Plasmid vector encoding human dysferlin cDNA	MM iPSCs	Restoration of dysferlin expression and rescue of defective membrane repair.	[49]
CRISPR/Cas9			
TALEN and CRISPR/Cas9	DMD iPSCs	Comparison of three correction methods: exon skipping, frameshifting, and exon knock-in; exon skipping was the most effective.	[50]
CRISPR/Cas9	iPSC-derived skeletal muscle fibers and CMs	Restoration of dystrophin by the CRISPR/Cas 9-mediated skipping of exons 45–55. Method applicable for 60% of DMD patient mutations.	[51]
CRISPR/Cpf1	iPSC-derived CMs	Cpf1-mediated genome editing restored dystrophin expression in CMs by skipping out-of-frame *DMD* exons or by correcting nonsense mutations. Reversal of dystrophy hallmarks after germline correction in a mouse model.	[52]
CRISPR/Cas9	DMD iPSC-derived CMs	Correction of dystrophin actin-binding domain mutations with expression of truncated dystrophin and rescue of CM features (contractility and calcium currents).	[53]
CRISPR/Cas9 with sgRNAs	DMD iPSC-derived 3D-engineered heart muscle	Correction of exon deletion, pseudoexons, point mutations and large duplication mutations; restoration of dystrophin expression and the corresponding mechanical force of contraction.	[54]
CRISPR/Cas9	DMD iPSC-derived myoblasts	Skipping of exon 45 and restoration of dystrophin expression.	[55]
CRISPR/Cas9	DMD iPSC-derived CMs	Correction of exon 44 deletion mutations using AAV9 encoding Cas9 and single-guide RNAs, leading to dystrophin restoration.	[56]
CRISPR/Cas3	DMD iPSC-derived myoblasts	Correction of exon 45 by Cas3; skipping and restoration of dystrophin was achieved, but the editing efficiency was lower than that of CRISPR/Cas9.	[57]
CRISPR/Cas9	DMD iPSC-derived myotubes and CMs	Correction of exon 52 deletion mutations using AAV9 intein-split Cas9 and a pair of guide RNAs, leading to dystrophin restoration and phenotypic rescue.	[58]
CRISPR/Cas9	DMD iPSC-derived myoblasts	Use of extracellular nanovesicles (NanoMEDIC) to deliver the CRISPR/Cas9 protein with a high transfection rate in different cell types and 90% of exon 45 being skipped in CMs.	[59]
CRISPR/Cas9-cytidine deaminase	DMD iPSCs	Use of a CRISPR-guided cytidine deaminase to induce exon 50 skipping and thus restore the ORF and dystrophin function.	[60]
CRISPR/Cas9	DMD iPSC-derived CMs	Use of an adenine base editor (ABE) to modify splice donor sites on the *DMD* gene, leading to skipping of the stop signal and correction of exon 51 deletions.	[61]
CRISPR/Cas9	LGMD R2 iPSCs and LGMD R3 iPSCs	Correction of point mutations leading to proper expression of dysferlin and α-SG proteins.	[62]
CRISPR/Cas9	LGMD R1 iPSC-derived myotubes	Correction of point mutations leading to proper expression of Calpain 3.	[63]
CRISPR/Cas9	DM1 iPSCs	Elimination of CTG pathological repeats.	[64]
CRISPR/Cas9	DM1 iPSC-derived myogenic cells	Removal of repeat expansion, thereby preventing nuclear focus formation and splicing alterations.	[65]
CRISPR/Cas9	DM1 iPSC-derived myoblasts	Removal of repeat expansion, thereby restoring myogenic capacity and nucleocytoplasmic distribution and preventing nuclear focus formation.	[66]
CRISPR/Cas9	DM1 iPSC-derived myogenic cells and myoblasts	Removal of repeat expansion, thereby preventing nuclear focus formation and splicing alterations.	[67]

Abbreviations: AAV, adeno-associated virus; ASO, antisense oligonucleotide; CK, creatine kinase; CM, cardiomyocyte; CRISPR/Cas9, clustered regularly interspersed short palindromic repeats/CRISPR-associated protein 9; DM1, myotonic dystrophy 1; DMD, Duchenne muscular dystrophy; IL, interleukin; LGMD, limb girdle muscular dystrophy; iPSC, induced pluripotent stem cell; MM, Miyoshi myopathy; SG, sarcoglycan; sgRNA, single-guide RNA.

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
