# Peer review of "Perspectives on hiPSC-Derived Muscle Cells as Drug Discovery Models for Muscular Dystrophies"

_ijms, 2021, doi:10.3390/ijms22179630_

Round 1

Reviewer 1 Report

The Authors summarized recent progresses in studies of muscular dystrophies by using disease-specific iPSCs, which is an interesting and timely topic. Overall, the manuscript covered recent important works, but the Reviewer suggests it should be addressed small concerns and revised with minor editing before publication.

1) L63-77: It would be better to explain separately DM1 and DMD.

2) "Principal models of MD": Why don't the Authors mention about iPSC-derived cardio myocytes, in addition to skeletal muscle cells?

3) Figure 1: Why don't the Authors mention about cardio myocytes?

4) It would be better to describe types of CRISPER/Cas clearly in body and in Table 1, as a number of variation have been developed.

5) Reference links should be added for each citation in Table 1. It helps readers to reach original studies easily.

6) It could help readers understand that iPSC studies in which drug screening was done were added to Table 1 or another table.

Author Response

The Authors summarized recent progresses in studies of muscular dystrophies by using disease-specific iPSCs, which is an interesting and timely topic. Overall, the manuscript covered recent important works, but the Reviewer suggests it should be addressed small concerns and revised with minor editing before publication.

Dear Reviewer, thank you very much for taking the time to review our paper, and for your positive comments. We modified the paper according to your suggestions. You may find below a point-by-point response to your concerns.

  • L63-77: It would be better to explain separately DM1 and DMD.

We discussed the available therapies for DMD and other MDs separately, as you mentioned.

  • "Principal models of MD": Why don't the Authors mention about iPSC-derived cardio myocytes, in addition to skeletal muscle cells?

We modified that section adding a paragraph about cardiomyocyte differentiation, as you recommended.

  • Figure 1: Why don't the Authors mention about cardio myocytes?

We modified the Figure (which is now Figure 2) adding cardiomyocyte differentiation passages.

  • It would be better to describe types of CRISPER/Cas clearly in body and in Table 1, as a number of variation have been developed.

We modified that section adding a discussion about different CRISPR/cas systems and modification done to Cas9, as you suggested.

  • Reference links should be added for each citation in Table 1. It helps readers to reach original studies easily.

We added reference links to references in the Table (which is now Table 2), as you recommended.

  •  It could help readers understand that iPSC studies in which drug screening was done were added to Table 1 or another table.

We added a Table (which is now Table 1) with relevant iPSC studies in which drug screening was done, as you suggested.

Reviewer 2 Report

Dear Authors,

this is a very interesting manuscript, summarizing the latest studies in the field of stem cell-based approach to discover efficient therapeutic strategies for muscular distrophies.  My ony minor concernis that the paper seems a little chaotic. Please add a figure/scheme summarizing described therapeutic approaches.

Author Response

Dear Authors,

this is a very interesting manuscript, summarizing the latest studies in the field of stem cell-based approach to discover efficient therapeutic strategies for muscular distrophies.  My ony minor concernis that the paper seems a little chaotic. Please add a figure/scheme summarizing described therapeutic approaches.

Dear Reviewer, thank you very much for taking the time to review our paper, and for your positive comments. We added an additional Figure (which is now Figure 1) that summarizes the therapeutic approach that we discuss in the paper.